# *Aspergillus welwitschiae* BK Isolate Ameliorates the Physicochemical Characteristics and Mineral Profile of Maize under Salt Stress

**DOI:** 10.3390/plants12081703

**Published:** 2023-04-19

**Authors:** Humaira Gul, Raid Ali, Mamoona Rauf, Muhammad Hamayun, Muhammad Arif, Sumera Afzal Khan, Zahida Parveen, Abdulwahed Fahad Alrefaei, In-Jung Lee

**Affiliations:** 1Department of Botany, Garden Campus, Abdul Wali Khan University Mardan, Khyber Pakhtunkhwa, Mardan 23200, Pakistan; gulhumaira@awkum.edu.pk (H.G.); Raidalikhan@gmail.com (R.A.); 2Department of Biotechnology, Garden Campus, Abdul Wali Khan University Mardan, Khyber Pakhtunkhwa, Mardan 23200, Pakistan; arif@awkum.edu.pk; 3Center of Biotechnology and Microbiology, University of Peshawar, Peshawar 25120, Pakistan; drsumaera@uop.edu.pk; 4Department of Biochemistry, Garden Campus, Abdul Wali Khan University Mardan, Khyber Pakhtunkhwa, Mardan 23200, Pakistan; zahida@awkum.edu.pk; 5Department of Zoology, College of Science, King Saud University, Riyadh 2455, Saudi Arabia; afrefaei@ksu.edu.sa; 6Department of Applied Biosciences, Kyungpook National University, Daegu 41566, Republic of Korea

**Keywords:** endophytic fungi, proline, salinity, maize, sodium, endogenous hormones, potassium

## Abstract

Abiotic stressors are global limiting constraints for plant growth and development. The most severe abiotic factor for plant growth suppression is salt. Among many field crops, maize is more vulnerable to salt, which inhibits the growth and development of plants and results in low productivity or even crop loss under extreme salinity. Consequently, comprehending the effects of salt stress on maize crop improvement, while retaining high productivity and applying mitigation strategies, is essential for achieving the long-term objective of sustainable food security. This study aimed to exploit the endophytic fungal microbe; *Aspergillus welwitschiae* BK isolate for the growth promotion of maize under severe salinity stress. Current findings showed that salt stress (200 mM) negatively affected chlorophyll a and b, total chlorophyll, and endogenous IAA, with enhanced values of chlorophyll a/b ratio, carotenoids, total protein, total sugars, total lipids, secondary metabolites (phenol, flavonoids, tannins), antioxidant enzyme activity (catalase, ascorbate peroxidase), proline content, and lipid peroxidation in maize plants. However, BK inoculation reversed the negative impact of salt stress by rebalancing the chlorophyll a/b ratio, carotenoids, total protein, total sugars, total lipids, secondary metabolites (phenol, flavonoids, tannins), antioxidant enzyme activity (catalase, ascorbate peroxidase), and proline content to optimal levels suitable for growth promotion and ameliorating salt stress in maize plants. Furthermore, maize plants inoculated with BK under salt stress had lower Na^+^, Cl^−^ concentrations, lower Na^+/^K^+^ and Na^+^/Ca^2+^ ratios, and higher N, P, Ca^2+^, K^+^, and Mg^2+^ content than non-inoculated plants. The BK isolate improved the salt tolerance by modulating physiochemical attributes, and the root-to-shoot translocation of ions and mineral elements, thereby rebalancing the Na^+^/K^+^, Na^+^/Ca^2+^ ratio of maize plants under salt stress.

## 1. Introduction

Salinity and its concomitant land-damaging effects are major environmental problems for agricultural production, impacting about 800 million hectares of land worldwide. Beyond soil salinity, irrigation using saline water, particularly in the low-lying coastal regions of several nations, has been highlighted as a key yield-limiting issue for boosting agricultural yields and productivity [1,2].

Plant salt stress responses are quite dynamic and complex and show variations depending on the origin, extent, inference, and impact of the plant stress. Salt stress also impacts seedling establishment and germination, and thus affects a plant’s life cycle, particularly in arid and semi-arid areas with little rainfall and high evapotranspiration. Soil salinity not only inhibits seed emergence and germination rates, but also drastically impairs field crop growth, development, and yield. High salt concentrations in the soil cause stomatal closure and damage to photosynthetic machinery and chlorophyll production [3,4].

Sodium ion (Na^+^) is the primary ion to cause salinity toxicity in most crop species, including maize. Plant responses to salt stress include decreased plant-available water induced by sudden osmotic stress as well as a Na^+^-specific component triggering biochemical disturbances [3,5]. In the latter stage of salt stress, abundant cytosolic Na^+^ components replace K^+^ to disrupt stomatal regulation [6], induce chloroplast deformation and dysfunction, and decrease enzyme activation and protein production [7]. Maize grown under salt stress has also shown a decrease in K^+^/Na^+^ ratios compared with non-salinized counterparts, as well as abnormalities in chloroplast structural components such as membrane, grana, and thylakoids, which are severely damaged at high salinity, concurrent with interrupted metabolism in the mesophyll cell [8,9]. Therefore, minimizing Na^+^ over-accumulation and maintaining K^+^/Na^+^ equilibrium (referred to as “homeostasis” in many articles) in the cytoplasm is critical for salt-stressed maize plants. Long-distance Na^+^ translocation from roots to shoots occurs via the xylem, and the shoots are the primary source of Na^+^ toxicity. As a result, techniques that promote Na^+^ removal from shoots while maintaining K^+^ have aided salt tolerance in plant species such as maize [10].

Additionally, reactive oxygen species (ROS) under salinity stress are overproduced in maize plants because of accelerated metabolic processes that are highly toxic and that oxidize macromolecules (nucleic acid, protein, carbohydrates, and lipids) while promoting cellular machinery dysfunction [11]. Plants have evolved an excellent antioxidant defense system that includes antioxidant enzymes such as superoxide dismutase (SOD), peroxidase (POD), ascorbate peroxidase (APX), and catalase (CAT). Moreover, phytohormones also participate in salt stress amelioration responses in plants [12].

In addition to this, halophytes have been shown to acquire the ability to cope with salt stress and maintain their growth normally, due to evolutionary processes and through morphological, physiological, and molecular plasticity in response to such stresses. Adaptive mechanisms include (i) osmolyte biogenesis and accumulation in governing turgor pressure, which prevents structural membrane injury, regularizes ionic homeostasis, and mediates ionic and water uptake and use efficiency; (ii) morpho-physiological flexibility and plasticity; (iii) enhanced photosynthetic and antioxidant potential; (iv) stress alleviation via phytohormonal biosynthesis and reshuffling; and (v) inherited flexibility in the altering of the gene expression controlling turgor pressure. Nonetheless, these adaptations are insufficient to confer resistance to globally ubiquitous salt stress as not all plant species are completely adapted [13].

Glycophytes, such as maize (*Zea mays* L.), are sensitive to high salt concentrations and suffer a severe reduction in yield, growth, and development when exposed to them [14]. Therefore, exploring new techniques for enhancing maize production and tolerance under salinity stress is both essential and challenging. The salinity threshold for maize in irrigation water is 1.1 dS m^−1^ and 1.7 dS m^−1^ in soil saturation extract [15]. However, tolerance varies across genotypes, salinity levels, and the kinds of salts present in water or soil [16,17]. Organic acids, osmoprotectants, hormones, and other attenuating compounds, such as salicylic acid [18], indol acetic acid (IAA) [19], and gibberellic acid [20], can improve a plant’s stress response. These hormones facilitate tissue development and growth, acting as intercellular communication mediators, and aiding in the translation of stress signals and homeostasis. Moreover, exogenous supplementation of exogenous Pi (H_2_PO_4_^−^) [21], putrescine [22], inulin [23], silicon [24], trehalose [25], paclobutrazol [26], glycine betaine [27], selenium [14], and melatonin [28], have also been shown to induce salt stress tolerance in maize. Nevertheless, the production and application of such compounds are laborious, expensive, and mostly hazardous to the environment and ecosystem. In addition to this, the effectiveness of these priming mediators, synthetic fertilizers, and growth regulators differs in so far as they mainly rely on environmental stress and plant species. Scientists around the world are searching for low-cost and rapidly implementable approaches to increase agricultural production and sustainability under stressful environments. The use of growth-promoting endophytic fungi is regarded as a favorable biological approach to proliferate agricultural outcomes and ensure the smooth operation of a successful agricultural system [29,30].

Endophytic fungi are symbiotically associated biota of living plant tissues that induce a symptomless disease in their hosts and are not host-specific. Endophytic fungi have captivated the attention of researchers over the last two decades as possible producers of biologically active compounds and growth-promoting hormones. Evidence has shown that endophytes have been co-evolving in their ability to synthesize compounds associated with those of their host plants, which might aid their hosts in developing their fitness and survival in an unfavorable environment [31,32]. In this context, the use of endophytic fungal isolates that are beneficial for plant growth to increase the production and sustainability of crops under salt stress appears to be a good strategy.

Endophytic symbionts found in global ecosystems are highly adapted to their environment. However, less than 5% of fungal species have been recognized so far, while at least 10 million microbial species are yet to be discovered [33]. Endophytic fungi have also been shown to be resistant to salt, drought, heat, osmotic stress, heavy metal stress, UV, and fungal pathogens [34,35,36,37]. Endophyte colonization with the host plant is thought to help the host plant adapt to biotic and abiotic stress components.

Considering previous knowledge about the role of endophytes on salinity stress tolerance in plants, the present study aimed to investigate the potential of BK’s association with maize and to elucidate the physiological responses of BK in the amelioration of salt stress in maize.

## 2. Results

### 2.1. Physiochemical Properties of the Soil

A soil mixture (sandy loam) from the Mardan district in Khyber Pakhtunkhwa (KP), Pakistan was collected for physicochemical analysis shown in Table 1.

### 2.2. Association of BK with Maize Roots and Root Hairs Abundance under Salt Stress

The morphological assessment of maize plants raised under salt stress with inoculation of BK showed an overall better growth response in comparison with non-inoculated plants under salt stress as well as control (Figure 1A).

Endophytic re-isolation and colony morphology of a BK re-isolated endophyte were undertaken to additionally validate the existence of endophytic fungus BK associated with the maize roots. The colony phenotype and microscopic morphology verified the extensive association of BK with the roots of BK-supplemented maize plants with and without salt stress (Figure 1B).

The symbiotic association of BK with maize roots in comparison with the control was further increased in maize plants cultivated in saline conditions, as clearly visible in Figure 1C, in terms of the blue coloration of lactophenol cotton blue, as well as the formation of the endophytic fungal hyphae. 

Microscopic examination also showed that the fungal hyphae were observed to be more abundantly generated under salt stress. The exaggerated and excessive lysogenic aerenchyma production in maize roots under saline conditions was reduced by supplementing the BK, as compared with the control (Figure 1C).

### 2.3. Effect of BK on Maize Growth under Salt Stress

The growth performance of maize plants grown in saline conditions revealed a significant (*p ≤* 0.05) decline in the average shoot length/plant (4.6 cm/9.1 cm (C)), root length/plant (3.0 cm/8.6 cm (C)), the total number of leaves from 3 plants/pot (12/24 (C)), fresh weight/plant ((0.5 g/1.04 g (C)), and dry weight/plant ((0.37 g/0.64 g (C)) as compared with unstressed control (C) maize plants. However, BK inoculation with maize under both the saline and normal conditions exhibited a significant (*p* ≤ 0.05) increase in examined parameters in comparison with the respective control maize plants.

BK endophytic association resulted in a significant (*p* ≤ 0.05) increase in shoot length/plant (8.5 cm/4.6 cm (SC)), root length/plant ((3.5 cm/3.0 cm (SC)), total number of leaves from 3 plants/pot (18/12 (SC)), fresh weight/plant ((1.76 g/0.5 g (SC)), and dry weight/plant (0.76 g/0.37 g (SC)) in maize plants grown under salt stress in comparison with non-inoculated stressed control (SC) plants (Figure 2A–E).

### 2.4. Effect of BK on Maize Photosynthetic Attributes under Salt Stress

Under salinity stress, maize plants showed significant (*p ≤* 0.05) inhibition in the production of different photosynthetic pigments such as chlorophyll a ((3.79 mg g^−1^ FW/5.7 mg g^−1^ FW (C)), chlorophyll b (0.95 mg/g FW/2.01 mg/g FW (C)), total chlorophyll (4.7 mg/g FW/7.78 mg g^−1^ FW (C)), and total carotenoid content (1.35 mg g^−1^ FW/2.0 mg g^−1^ FW (C)), while promoting chlorophyll a/b ratio (4.9/2.7 (C)), as compared with the unstressed control (C) maize plants.

BK inoculation to maize plants under both salt stress and normal conditions resulted in a significant increase in the production of chlorophyll a, chlorophyll b, total chlorophyll, and carotenoid content compared with non-inoculated respective control maize plants.

BK inoculation to maize plants under salt, stress showed significant (*p ≤* 0.05) promotion in the synthesis of the chlorophyll a, (4.315 mg g^−1^ FW/3.79 mg g^−1^ FW (SC)), chlorophyll b (1.94 mg g^−1^ FW/0.95 mg g^−1^. FW (SC)), total chlorophyll (6.15 mg g^−1^ FW/4.7 mg g^−1^ FW (SC)), and total carotenoids (2.50 mg g^−1^ FW/1.35 mg g^−1^ FW (SC)), with a reduction in the chlorophyll a/b ratio (1.75/4.0 (SC)) in maize plants grown under salt stress in comparison with non-inoculated stressed control (SC) maize plants (Figure 3A–E).

### 2.5. Effect of BK on Maize Primary Metabolites under Salt Stress

Under salinity stress, maize plants showed significant (*p ≤* 0.05) reduction in growth-related primary metabolites such as total soluble sugars (31.8 mg g^−1^ FW/37.7 mg g^−1^ FW (C)), total proteins (0.48 mg g^−1^ FW/0.75 mg g^−1^ FW (C)), and total lipids (0.15 mg g^−1^ FW/0.21 mg g^−1^ FW (C)), as compared with unstressed control (C) maize plants.

BK inoculation to maize plants under both salt stress and normal conditions resulted in a significant increase in the production of primary metabolites (total soluble sugars, total proteins, total lipids) as compared with non-inoculated respective control (C) maize plants.

BK inoculation to maize plants under salt, stress showed significant (*p ≤* 0.05) promotion in the synthesis of primary metabolites, such as total sugars (55.16 mg g^−1^ FW/31.8 mg g^−1^ FW (SC)), total proteins (1.41 mg g^−1^ FW/0.48 mg g^−1^ FW (SC)), and total lipids (0.26 mg g^−1^ FW/0.15 mg g^−1^ FW (SC)), in maize plants grown under salt stress in comparison with non-inoculated stressed control (SC) maize plants (Figure 4A–C).

### 2.6. Effect of BK on Osmolyte Production in Maize under Salt Stress

Under salinity stress, maize plants showed significant (*p ≤* 0.05) promotion in the production and accumulation of proline (osmolyte) (0.10 µg·g^−1^ FW/0.03 µg·g^−1^ FW (C)), as compared with unstressed control (C) maize plants.

BK inoculation to maize plants under both salt stress and normal conditions resulted in a significant increase in the production of proline, as compared with non-inoculated respective control maize plants.

BK inoculation to maize plants under salt, stress showed significant (*p ≤* 0.05) promotion in synthesis and accumulation of proline content (0.21 µg·g^−1^ FW/0.10 µg·g^−1^ FW (SC)) in maize plants grown under salt stress in comparison with non-inoculated stress maize plants (Figure 4D).

### 2.7. Effect of BK on Maize Secondary Metabolites under Salt Stress

The maize plants exposed to salt stress showed a significant (*p ≤* 0.05) increase in secondary products of metabolism, such as total phenolic contents (70.9 µg·g^−1^ FW/40.5 µg·g^−1^ FW (C)), total flavonoids (1.75 µg·g^−1^ FW/1.54 µg·g^−1^ FW (C)) and total tannins (170.2 µg·g^−1^ FW/84.4 µg/g FW (C)), compared with control (C) plants. 

BK inoculation to maize plants under both salt stress and normal conditions resulted in a significant increase in the production of secondary metabolites as compared with non-inoculated respective control maize plants.

BK endophytic association significantly increased secondary metabolites, such as total phenols (75.6 µg·g^−1^ FW/70.9 µg·g^−1^ FW (SC)) and total flavonoids (2.65 µg·g^−1^ FW/1.75 µg·g^−1^ FW (SC)), however total tannins (117.5 µg·g^−1^ FW/170 µg·g^−1^ FW (SC)) were reduced in maize plants grown under salt stress in comparison with non-inoculated stressed control (SC) plants (Figure 5A–C).

### 2.8. Effect of BK on Endogenous Level of Growth-Promoting Hormonal Content in Maize under Salt Stress

Under salinity stress, maize plants showed a significant (*p ≤* 0.05) reduction in the endogenous level of growth-promoting hormone (indole acetic acid—IAA) (0.36 mg/g FW/1.51 mg/g FW (C)), as compared with unstressed control (C) maize plants.

However, BK inoculation to maize plants under both salt stress and normal conditions resulted in a significant increase in the production of IAA, as compared with non-inoculated respective control maize plants.

BK inoculation to maize plants grown under salt stress, showed significant (*p ≤* 0.05) promotion in IAA content (0.07 µg·g^−1^ FW/0.36 µg·g^−1^ FW (SC)), in maize plants grown under salt stress in comparison with non-inoculated stressed control (SC) maize plants (Figure 5D).

### 2.9. Effect of BK on MDA Content and Antioxidant Enzymes of Maize under Salt Stress

Adversative effects of salt stress were observed in maize plants in terms of a significant (*p ≤* 0.05) increase in the MDA content (0.043 nmol·g^−1^ FW/0.025 nmol·g^−1^ FW (C)), as well as antioxidant enzyme activities such as catalase (CAT) (307.8 units·g^−1^ FW/185.2 units·g^−1^ FW (C)), ascorbate peroxidase (APX) (142.8 units·g^−1^ FW/105.4 units·g^−1^ FW (C)), as compared to non-inoculated control (C) plants (Figure 6). 

However, BK inoculation to maize plants under both salt stress and normal conditions resulted in a significant decrease in the production of MDA content, as compared with non-inoculated respective control maize plants. While antioxidant enzyme activities CAT and APX were further increased by BK inoculation to maize plants under both the salt stress compared with the control.

BK endophytic association significantly reduced the MDA content (0.022 nmol·g^−1^ FW/0.043 nmol·g^−1^ FW (SC)), while CAT activity (574.7 units·g^−1^ FW/307.8 units·g^−1^ FW (SC)), and APX activity (269.1 units/g FW/142.8 units·g^−1^ FW (SC)) were increased, in maize plants grown under salt stress in comparison to non-inoculated stressed control (SC) maize plants (Figure 6A–C).

### 2.10. Effect of BK on Ionic Contents and Mineral Elements of Maize under Salt Stress

Current findings show the ionic status of corn plants under salt stress conditions and the plants show a significant increase (*p* < 0.05) in Na^+^ ions (70.8 mg g^−1^ DW/14.1 mg g^−1^ DW (C)) and Cl^−^ ions (55.9 mg g^−1^ DW/11.7 mg g^−1^ DW (C)) and decrease in K^+^ (24.9 mg g^−1^ DW/33.1 mg g^−1^ DW (C)), Ca^+^ (10.8 mg g^−1^ DW/17.3 mg g^−1^ DW (C)), and Mg^2+^ (9.4 mg g^−1^ DW/15.8 mg g^−1^ DW (C)), compared with non-stressed and non-inoculated control (C) plants (Figure 7A). 

Mineral elements, such as phosphorus (3.6 mg g^−1^ DW/18.3 mg g^−1^ DW (C)), and nitrogen contents (26.3 mg g^−1^ DW/50.5 mg g^−1^ DW (C)), were also decreased by salt stress compared with non-stressed and non-inoculated control (C) plants.

However, BK endophytic association decreased the ionic contents of Na^+^ ions (21.0 mg g^−1^ DW/70.8 mg g^−1^ DW (SC)) and Cl^−^ ions (12.3 mg g^−1^ DW/55.9 mg g^−1^ DW (SC)), while increasing K^+^ (51.5 mg g^−1^ DW/24.0 mg g^−1^ DW (SC)), Ca^+^ (17.1 mg g^−1^ DW/10.8 mg g^−1^ DW (SC)), and Mg^2+^ (20.1 mg g^−1^ DW/9.4 mg g^−1^ DW (SC)), in maize plants grown under salt stress in comparison with non-inoculated stressed control (SC) maize plants (Figure 7A). 

Mineral elements, phosphorus (19.6 mg g^−1^ DW/3.6 mg g^−1^ DW (SC)), and nitrogen contents (56.3 mg g^−1^ DW/26.3.5 mg g^−1^ DW (SC)) were also increased by the BK inoculation in maize plants grown under salt stress in comparison with non-inoculated stressed control (SC) maize plants (Figure 7A). 

The BK association also decreased the Na^+^/K^+^ (0.64/4.1 (SC)), and Na^+^/Ca^2+^ ratios (1.40/6.8 (SC)) in maize plants grown under salt stress in comparison with non-inoculated stressed control (SC) maize plants (Figure 7B).

## 3. Discussion

The exploitation of fungal endophytes to salt-sensitive plants such as maize, in a saline environment, is believed to be an attractive strategy for the restoration of the sustainable growth of crops under stressful, saline environments for plant growers across the world [38,39]. In the current study, endophytic fungal isolate BK induced salt tolerance in maize under saline stress conditions that imposed a substantial reduction in the overall growth and biomass of maize without BK inoculation. BK inoculation reverses the negative effects of salt on maize plants in terms of growth promotion. Excessive salinity inhibits plant growth by excessive absorption of Na^+^ and Cl^−^ and causes disturbances in ion balance and physiological disorders. By reducing water absorption, reducing turgor, increasing stomatal closure, and reducing photosynthetic activity in different plant species, it causes osmotic stress and reduced plant growth. Roots can exclude and/or prevent the uptake of potentially toxic ions through morphological and molecular plasticity [24,40]. In the present study, maize plants exposed to salt stress showed a significant decrease in stem length, root length, number of leaves, and fresh and dry biomass compared with control plants.

Excess Na^+^ causes certain ion toxicity, so it is important to control the removal and transport of Na+. The fewer other nutrients, such as potassium and calcium, are absorbed, the higher the sodium and chlorine levels, leading to a nutritional imbalance. Plants can produce ROS as a result of salinity stress. Increased ROS causes further lipid peroxidation, protein degradation, and DNA mutation [41]. Salinity affects the number of leaves, leaf area, shoot, and root dry weight, resulting in reduced yields. Premature leaf senescence and defoliation result in a decrease in leaf number under salinity and water stress. Plants’ fresh weights and dry weights were observed to be reduced because of a proportional increase in Na^+^ concentration. In addition to the harmful effects of NaCl, a higher salt concentration lowers the water potential in the medium, preventing water absorption and hence limiting plant growth [42]. Under harsh environmental conditions, endophytic fungal associations keep plants fit and healthy.

Previous reports have shown that inoculation of endophytic fungus (*P. formosusto)* in cucumber plants grown under salt stress dramatically increased branch length and improved growth parameters compared with the non-inoculated plants under salt stress. Consistent with the previous reports, the current study also revealed the positive influence of BK inoculation, which promoted growth attributes such as shoot length, root length, number of leaves, and fresh and dry biomass of maize plants under salt stress conditions.

The present study also showed that corn exposed to salt stress had significantly lower levels of chlorophyll a, chlorophyll b, and total chlorophyll and an increased ratio of chlorophyll a/b and carotenoids. Reduced production or increased breakdown of chlorophyll molecules under salinity stress limits photosynthetic activity. The toxicity of Na^+^ or salt-induced oxidative damage triggers the disintegration of the chloroplast ultrastructure. Decreased photosynthetic pigment, stomatal conductance, impaired enzyme activity, and reduced photosynthetic activity are all important characteristics that limit maize plants’ carbon fixation capacity under salt-stress conditions. However, recently [39] found that a growth-promoting endophytic fungus (*Stemphylium lycopersici*) ameliorated salt stress tolerance in maize by balancing the ionic and metabolic status of plants. Consistent with the previous reports, the present research also revealed the role of *c* in increasing the levels of chlorophyll a, chlorophyll b, total chlorophyll, chlorophyll a/b ratio, and carotenoids under both normal and stressed conditions. As previously reported, a higher photosynthetic rate is linked to an increased chlorophyll concentration in endophyte-associated plants and previously been found that the endophyte *P. indica* efficiently reduces drought-induced chlorophyll content reductions by keeping the photosynthetic rate constant [43].

Proline levels in stressed plants can help to maintain membrane integrity, reduce lipid membrane oxidation, protect and stabilize ROS-scavenging enzymes, and play a role in subcellular structure stabilization, free radical scavenging, and redox potential buffering. Maize plants grown under salt stress had a considerable increase in total proline compared with control plants. Proline concentrations have been observed to rise to 100 times normal in salinity conditions, accounting for the amino acid pool of up to 80%. Proline content in plant tissues acts as a stress-induced marker indicating protein damage in the cell. Nevertheless, proline production during stress maintains the cellular balance for cytosol’s osmotic potential with that of the vacuole and the external environment [44], *Z. mays* L. [45], and *O. sativa* [46]. The PGPR (*A. chrocoocum* A101) significantly improved proline accumulation, resulting in increased water intake, water use efficiency, and photosynthetic efficiency [47].

Plant metabolic activities are disrupted by changes in total sugar levels under abiotic stress conditions. The accumulation of total soluble sugars, such as glucose, sucrose, dextrins, and maltose, is a survival strategy through osmotic readjustment employed by salt-stressed plants, as these sugars play a role as an osmoprotectant and as carbon storage compounds. Moreover, sugar also allows for the avoidance of crystallization of the molecules within the cell, hence limiting cellular structure damage. The quantities of total soluble sugars in leaves appear to be less influenced by salinity than in roots. Under salinity stress, protein content is reduced due to nitrogen metabolism disturbances or nitrate absorption suppression [48,49].

In the present research, BK supplementation in maize plants under salt stress also demonstrated a high level of total sugars, total proteins, and total lipids contents. This is linked to the growth regulators’ favorable morphological and physiological effects on the host plant. Increased vegetative growth characteristics, such as shoot length, root length, fresh biomass, and dry biomass may also be justified by the beneficial effects of BK on photosynthetic pigments and stimulating photosynthesis machinery, as indicated in the data. Induced signaling components for the phytohormone production pathways that aid growth promotion under salt stress may also be caused by the sugar buildup associated with BK supplementation in maize plants. Salinity has also been shown to increase the generation of reactive oxygen species (ROS) in cells, which can destroy essential cell components such as proteins, lipids, and nucleic acids. However, lipid and protein concentrations in plants decrease dramatically at the 200 mM salinity level [50] and suppress the gene expression responsible for controlling oxidative stress responses in *Z. mays* L. [45].

Endophytic fungi seem promising in their ability to enhance plant growth and nutrient absorption, as well as to alleviate the impacts of salt stress. Long-term solutions to ensure the security of the world’s food supply can be solved by the application of endophytic fungi [51]. A growth-promoting fungus, *A. aculeatus,* increases salt-stress tolerance, metabolite activities, and fodder quality in perennial ryegrass [52], while *T. harzianum* reduces salt stress in cucumber [53], and *P. indica* reduces salinity stress in *M. truncatula* [54]. Likewise, the current study also revealed that maize plants exposed to salt stress had a significantly higher level of soluble sugars, proteins, and lipids compared with the control. However, plants associated with BK also showed higher total sugar, proteins, and lipid levels under high salinity than non-saline endophyte-associated plants, which is suggestive of the survival activity of maize plants under salt stress. Many plants have different kinds, compositions, and levels of phenolic compounds produced because of environmental and genetic variations. Flavonoids, glutathione, ascorbic acid, carotenoids, and tocopherol all operate in the cell to maintain membrane integrity [55]. Flavonoid biosynthesis provides plants with a large range of secondary metabolites, such as flavonols, anthocyanins, and condensed tannins (CT), that are polyphenolic compounds and are synthesized by higher plants by their perceiving of external signals from environmental perturbances, as well as through internal metabolic cues. Tannins act as plant protectors against pathogens, pests, and diseases and regulate seed permeability and dormancy [56].

In the present research, maize plants under salt stress have shown a considerable increase in total phenols, flavonoids, and tannins. The polyphenol content is known to increase as saline levels increased. Flavonoids, particularly iso-flavonoids, play an important role in abiotic and biotic defensive mechanisms in several leguminous plants. Much research has corroborated the occurrence of increased amounts of anthocyanins and flavonoids in various plant species after salt stress was applied. Tannins are polymeric flavanols made up of monomeric components such as flavan-3-ols and flavan-3-4-diols that combine to generate polymeric flavanols. The high amounts of tannins in combination with the high levels of fiber in the leaves may harm their quality [57]. In the present study, BK association in maize plants boosted total phenols, flavonoids, and tannin in both normal and stressed conditions compared with non-saline endophyte-associated plants. [58] found that increased proline and phenol accumulation in *B. subtilis*-associated chickpea plants resulted in stress resistance. Proline mitigates the harmful effects of salinity by enabling the ROS scavengers to shield proteins and other biomolecules, in addition to their important roles in maintaining cell water balance. [59] also discovered increased total flavonoids in fungal-infested wine grapes. Tannin has been postulated to have a role in pathogenic and mutualistic interactions between plants and microbes, as well as plant responses to abiotic stressors. 

Salinity-induced oxidative stress and plant susceptibility are frequently measured using lipid peroxidation, a non-enzymatic autoxidation process caused by ROS [60]. Lipid peroxidation is the oxidative breakdown of lipids with any number of carbon–carbon double bonds. Lipid peroxidation is a well-known cellular damage mechanism in both plants and animals and is used to identify oxidative stress in cells and tissues. When exposed to air, lipid peroxides decompose, producing a complex chain of molecules, including reactive carbonyl compounds. Lipid peroxidation is involved in many normal developmental processes, including the generation of flavor and odor volatiles, the formation of molecules with growth-regulator-like functions, and senescence. The maize plants under salt stress have shown an increase in lipid peroxidation, catalase, and ascorbate peroxidase activity compared with the control. This is usually assessed in terms of MDA levels, which are prevalent lipid peroxidation end products. MDA concentrations rise with increasing salinity in halophytes. According to [61], a considerably increased activity of the catalase enzyme is attributable to the H_2_O_2_ overproduction upon salt stress that results in a significant reduction in the growth of plants. By neutralizing reactive oxygen species, the ascorbate peroxidase enzyme protects plants from osmotic stress (ROS). In rice, salt stress boosts the activity of an antioxidant enzyme called APX [62].

During the study, the use of the endophyte *A. welwitschiae* boosted lipid peroxidation, catalase, and ascorbate peroxidase activity in both normal and stressed conditions as compared with non-saline endophyte-associated plants. Under stress, changes in lipid peroxidation serve as a measure of the severity of oxidative damage. To examine the NaCl-induced oxidative damage in rice, lipid peroxidation is used as a measure of biological membrane disintegration. With *P. indica* colonized rice, endophytes dramatically reduced lipid peroxidation at all NaCl concentrations [63]. Lipid peroxidation was reduced in plants treated with NaCl in combination with AMF. One explanation may be that antioxidant enzymes can scavenge ROS before they react with membrane lipids and reduce lipid peroxidation. APX activity increased with increasing NaCl levels in rice inoculated with *P. indica,* and CAT activity increased with increasing NaCl levels. Similar to shoot catalase activity in the control group, control root catalase activity increased NaCl by up to 200 m before being reduced to 300 mM [63]. D Lipid peroxidation was reduced in plants treated with NaCl in combination with AMF. One explanation may be that antioxidant enzymes can scavenge ROS before they react with membrane lipids and reduce lipid peroxidation. The APX and CAT enzyme activity is accelerated due to the increased NaCl levels in rice inoculated with *P. indica* [63]. Because increased antioxidant enzyme activity reduces the risk of damage by ROS over production and accumulation, *P. indica* may protect the plant from oxidative damage.

Plants use phytohormone induction as one of their abiotic stress-resistance mechanisms, which enhances plant growth and output in stressful environments. The maize plants under salt stress have shown a significant drop in endogenous IAA levels in comparison with the control. Increased abscisic acid levels were found at the cost of indole acetic acid in maize [64]. This change could contribute to stomatal closure to prevent water loss due to osmotic stress caused by salinity. Root tips are the first to recognize reduced water availability in a saline environment due to the osmotic effect, sending a signal to shoots to adjust whole-plant metabolism [65]. The endophyte *A. welwitschiae* inoculation has increased the endogenous IAA levels in maize plants in normal growth conditions as well as under salt stress. Previously, rice growth had been improved by inoculating the IAA-producing fungal and bacterial endophytes under drought, salinity, and high-temperature stress. 

High salts (Na^+^ and Cl^−^) in the soil reduce nutrient availability by obstructing nutrient absorption, transport, or distribution inside the plant. The overaccumulation of Na^+^ and Cl^−^ is also followed by a huge fall in cytosolic K^+^, which further participates in the destructive metabolic activities of a cell. Sodium and chloride ion toxicity can produce ROS, which can damage cellular functioning [66]. A good natural balance of nutrients should be maintained by avoiding excessive intake of salt ions (both macro and micro). Rising Na^+^ concentrations increase the Na^+^/K^+^ ratio by inhibiting cytosolic activities, which affects respiration and photosynthesis [67]. Treatments with NaCl and Na_2_CO_3_ raised the K^+^/Na^+^ ratio by 12 and 17 times, respectively [68]. Increased Na^+^ and K^+^ competition in root and leaf tissues affects cellular metabolism. High Na^+^/K^+^ ratios in the cytosol interfere with stomatal movement, protein synthesis, turgor maintenance, and photosynthesis [69]. Mycorrhizal plants at high salinity have been observed to have decreased Na^+^ uptake, demonstrating that the AM fungus regulates Na+ absorption when it becomes hazardous to the plant. The plant’s cell membranes and water potential might both be harmed by the increased ionic flow. A rise in the Na^+^/K^+^ ratio in the root and shoot tissue of rice plants under salt stress is a noteworthy effect of salt. In contrast, *P. indica* inoculation increased the K^+^ level and decreased the Na+ concentration in rice plants under salt stress [70]. According to this study, maize plants exposed to 200 mM salinity stress showed considerably greater sodium and potassium concentrations, as well as Na^+^/K^+^ and Na^+^/Ca^2+^ ratios, than non-saline control plants., however, this negative effect of salinity was reversed by the endophytic application.

The high sodium/potassium, sodium/calcium, and sodium/magnesium ratios in the total plant confirm that sodium-mediated ionic transport (magnesium, potassium, calcium) may disrupt plant metabolism, resulting in reduced growth in saline conditions. Treatment of rice seedlings with endophyte strain D1 had positive results in both normal and saline conditions, decreasing Na^+^ absorption [68]. During the study, the endophyte BK elevated the sodium and potassium concentrations as well as the Na^+^/K^+^ and Na^+^/Ca^2+^ ratios in both normal and stressed conditions. The impact of salinity on potassium and sodium feeding can be reversed by fungal association, as under saline conditions, mycorrhizal colonization can boost potassium retention while preventing sodium transfer to shoot tissues. The synthesis and storage of polyphosphate, as well as various cations, particularly potassium, may influence sodium uptake. While photosynthesis and other metabolic activities of the cell rely heavily on ionic equilibrium. The soil supplemented with AMF endophytes was particularly efficient in reducing the negative impact of salt stress on beans’ ionic balance, mitigating the K^+^ deficit, and lowering the ratios of Na^+^/K^+^ and Na^+^/Ca^2+^ [66]. Potassium competes with sodium in saline conditions. When Na^+^ ions reach a cell’s plasma membrane in saline soil, the cell’s plasma membrane depolarizes and the K^+ outward^ rectifier channels open and the cell loses K^+^ [71]. More physiological functions show that calcium aids in solute mobility, stomatal control, molecular communication for cell defense systems, and cell repair under stress. Salinization also renders P unavailable to plants due to precipitation with other cations such as Ca^2+^, Mg^2+^, and Zn^2+^ depending on the pH of the soil environment, resulting in salt-induced P deficiency in plants. In the present study, the maize plants exposed to salt stress also showed a considerable decrease in ionic concentrations, compared with the control. Potassium functions as a secondary messenger in the regulation of abiotic stress signaling pathways and the promotion of K^+^/Na^+^ selectivity. Furthermore, calcium has been demonstrated to alleviate salt stress by promoting the selective absorption of potassium in plants exposed to greater sodium concentrations, as well as allowing plants to modify their osmotic balance via increasing ion uptake and aquaporin regulation. Salt stress inhibits nitrogen uptake and translocation in addition to potassium and calcium, resulting in lower nitrogen levels in various maize organs. Because many of these compatible solutes are nitrogen-containing substances, such as amino acids, amides, and betaines, nitrogen metabolism is critical under stressful situations. Furthermore, Mg^2+^ deficiency also affects plant growth due to salt. Salt tolerance was given by the fungal isolates investigated via increasing K^+^ accumulation and decreasing Na^+^ concentration. Endophytic fungi, have beneficial impacts on ion homeostasis, allowing plants to conserve strong K^+^ uptake while accumulating little Na^+^ during salt stress [52,54].

Previously, it was also known that, even at high salinity levels, K^+^ uptake expanded in mycorrhizal plant shoot tissues (9.5 dS m^−1^). Many studies have found that fungal association improves plant nutrient uptake (Mg^2+^, P, N) in salt stress conditions. This increases the proportion of K^+^/Na^+^ in the roots and shoots of plants. During mutualistic symbiosis, higher N uptake by the host plant may have helped to minimize the negative impact of the Na^+^ ions upon NaCl-mediated stress [72]. In the current research, the endophyte BK also raised the potassium, calcium, magnesium, phosphorus, and nitrogen levels under normal, as well as saline conditions.

## 4. Materials and Methods

### 4.1. A. welwitschiae BK Purification and Spore Suspension Preparation

Endophytic fungal isolate *A. welwitschiae* BK was isolated and identified in the Plant–Microbe Interaction (PMI) Laboratory at Abdul Wali Khan University Mardan (AWKUM), from the host plants *Chlorophytum comosum* as reported by [73]. BK was refreshed by being grown in a 50 mL liquid media (peptone 1%, C_6_H_12_O_6_ 1%, MgSO_4_·7H_2_O 0.05%, KCl 0.05%, FeSO_4_·7H_2_O 0.001%, pH 7.3) using 250 mL conical flasks, being kept at 30ºC and at 120 rpm in a shaking incubator for up to one week. The spore suspension was attained at a final concentration of 5 × 10^7^ spores/mL, which was maintained and employed for further experiments in this study.

### 4.2. Microscopy of the Endophytic Fungus BK

The mycelium of BK was visualized for morphological characterization under a light microscope (Binocular NSL-CX23 Olympus, Tokyo, Japan) by using the lactophenol cotton blue (staining agent). Microscopy was performed first at low magnification and then at high magnification, 40× and 100×.

### 4.3. Endophytic Fungal BK Inoculation to Maize Plants under Salt Stress

To investigate the performance of BK inoculation with maize plants, an endophytic fungus–plant association bioassay was performed. *Zea mays* (Var. Gulibathi) seeds received from the Agricultural Research Institute (ARI), Ternab, Peshawar, were sterilized with 0.1% mercury chloride for 60 s and then washed three times with distilled water. Sterilized seeds were germinated in glass Petri dishes and, 4 days after germination, uniformly germinated seedlings were transferred to soil pots pre-mixed with fungal biomass. Liquid spore suspension (5 × 10^7^ spores/mL) of BK was inoculated for 7 consecutive days (1 mL per seedling and 3 seedlings per pot).

Controls were irrigated with the same amount of irrigation water. The experimental set consisted of 12 plastic pots/treatments (diameter 8.5 cm, depth 12.5 cm), and 500 g of sterile sandy loam soil that was analyzed for physiochemical properties as shown in Table 1.

Previous reports have shown that plant maize is relatively sensitive to salinity stress, with growth inhibition of up to 200 mM NaCl [74]. However, the present study was simplified to test the role of ameliorating salt stress at 200 mM NaCl concentration by inoculating a growth-promoting endophytic fungus (*A. welwitschia* BK). For this purpose, 5-day-old maize plants, pre-inoculated with the BK isolate, were irrigated with a 2 mL solution of 200 mM sodium chloride (NaCl) at alternative days till plants were 20 days old, followed by a recovery period of 10 days. Control plants were supplied with an equal volume of tap water.

The experimental design was completely randomized with 4 treatment groups and repeated 3 times.

**Set-I**: Irrigation of maize plants with tap water (0 mM NaCl);

**Set-II**: Irrigation of maize plants with 200 mM NaCl;

**Set-III**: Maize plants inoculated with spores of BK and irrigation of plants with tap water (0 mM NaCl);

**Set-IV**: Maize plants inoculated with spores of BK and irrigation of plants with 200 mM NaCl.

To observe initial vegetative growth, experiments were undertaken for 35 days and maize plants were harvested. The roots and shoots were carefully detached and cleaned with running water to eliminate dirt and soil particles and to allow for the measurement of fresh weight, while the dry weight of the same pieces was measured after drying in an oven at 60 °C for 72 h.

### 4.4. Assessment for Photosynthetic Pigments in Maize Plants Inoculated with BK under Salt Stress

The quantification of the photosynthetic pigments (chlorophyll a, chlorophyll b, and carotenoid content) in fresh leaves was performed as described by [75]. Fresh tissues from leaf samples (0.3 g) were crushed in 80% acetone (3 mL) and centrifuged for 5 min at 1000 rpm. After centrifuging the pellet of plant material three times, the supernatant was pooled, and a final volume of 7 mL was attained by adding 80% acetone. The absorbance was recorded through a UV/Vis spectrophotometer (PerkinElmer Inc., Waltham, MA, USA) at 663 nm (chlorophyll-a), 645 nm (chlorophyll-b), 480 nm, and 510 nm (carotenoids).

### 4.5. Quantification of Primary Metabolites in Maize Plants Inoculated with BK under Salt Stress

Total soluble sugars were estimated according to the method described earlier [76]. An amount of 0.5 g of fresh leaves was crushed in 10 mL of H_2_O, and centrifuged for 5 min at 3000 rpm and 0.1 mL of supernatant was mixed with 1 mL of phenol (80%), the sample was then incubated for 10 min at a time. After adding 5 mL of concentrated H_2_SO_4_, the mixture was incubated for one hour. At 485 nm, the optical density of solutions was measured, and the contents of the soluble sugar were expressed as mg g^−1^ FW. 

The total protein content in maize plant tissues was estimated as described earlier [77]. Ice-cold phosphate buffer (pH = 7, 0.1 M potassium phosphate) (5 mL) was added to crush the leaf tissue (0.3 g) and incubated in the freezer for 20 min with subsequent centrifugation at 12,000 rpm. About 4.8 mL of buffer was added to 0.2 mL of the extract for dilution, followed by the addition of Bradford reagent (5 mL) to 0.1 mL of diluted extract. The optical density was recorded at 595 nm.

The lipid content of maize leaves was determined as described by [78]. Chloroform: methanol (2:1 *v*/*v*) was used to grind a leaf sample (0.3 g), followed by the addition of 0.73% NaCl (0.8 mL). Three layers containing various chemicals were separated, with the lower layer containing lipids extracted by the addition of sulfuric acid (0.1 mL). The mixture was shaken and heated at 100 °C for 10 min with subsequent cooling and adding 2.4 mL of vanillin reagent. The absorbance was measured at 490 nm.

### 4.6. Quantification of Secondary Metabolites in Maize Plants Inoculated with BK under Salt Stress

Total phenolics extraction and estimation were performed as described earlier [79]. The maize leaf tissue (0.5 g) was ground in dH_2_O (10 mL), the extract was separated into a glass tube and the total volume was adjusted up to 3 mL, with the subsequent addition of Folin Ciocalteau reagent (0.5 mL) and Na_2_CO_3_ (2 mL). Catechol (CellMark AB, Göteborg, Sweden) was used to generate a standard curve with known concentrations. The absorbance was measured at 650 nm.

Total flavonoids were estimated as described earlier [80], using the maize leaf tissue (5 g) that was macerated in 50 mL of 80% ethanol. After 24 h of incubation periods, the extract was centrifuged at 10,000 rpm for 15 min, a volume of 250 µL of the extract was combined with 1.25 µL of dH_2_O and 7 µL of a 5% NaNO_2_ solution followed by a subsequent incubation and the addition of 150 µL of 10% AlCl_3_.H_2_O. The reaction was further incubated for 6 min, followed by the addition of 500 µL of 1M NaOH and 275 µL dH_2_O. Known concentrations of quercetin (TargetMol, Boston, MA, USA) were used to plot the calibration curve, whereas ethanol (80%) was used as a blank, and absorbance was measured at 415 nm.

Total tannins were quantified as described earlier [81], in maize leaf tissues (200 mg) that were homogenized in 70% acetone and incubated in a shaking incubator for 6 h. The solutions were read at 725 nm after the addition of 0.5 mL folin and 2.5 mL 20% NaCO_3_. Known concentrations of tannic acid (Sigma-Aldrich (St. Louis, MO, USA) was used to generate a standard curve.

### 4.7. Quantification of Osmolytes in Maize Plants Inoculated with BK under Salt Stress

Proline is the most common endogenous osmolyte accumulated under various abiotic stresses, including salinity, and was also quantified in maize plants inoculated with BK under salt stress. In [82] the method was followed to extract and estimate proline content in maize leaf tissue (0.5 g) that had been powdered in a 10 mL sulfosalicylic acid (3%) solution and centrifuged for 5 min at 3000 rpm, while 2 mL of glacial acetic acid and 2 mL of freshly made ninhydrin reagent were added to 2 mL of supernatant. The mixture was heated to 100 °C for an hour, with subsequent cooling and the addition of 4 mL of toluene. The absorbance was measured at 520 nm and a standard curve was generated using known concentrations of proline (SENOVA Technology Co., Ltd., Cambridge, UK).

### 4.8. Quantification of MDA Content and Antioxidant Enzyme Activities in Maize Plants Inoculated with BK under Salt Stress

MDA content was measured as described earlier [83], by taking 0.5 g of the finely crushed leaf tissue sample, homogenized with 3 mL of 0.6% thiobarbituric acid (TBA), centrifuged (10 min; 12,000 rpm), boiled in a water bath at 100 °C for 10 min, cooled, and centrifuged again (12,000 rpm; 10 min). Finally, absorbance was measured at 532, 450, and 600 nm.

Catalase activity was quantified through H_2_O_2_ cleavage as described earlier [84], by taking 0.1 mL of supernatant and adding a phosphate buffer of 2.6 mL pH 7, 0.1 mM EDTA and 400 microliters of 3% H_2_O_2_. The absorbance was measured at 240 nm. For the estimation of ascorbate peroxidase, the protocol of [85], was used by mixing 0.2 mL of leaf extracts with 600 microliters of PBS (50 mM, pH 7.0), 100 microliters of ascorbic acid (0.5 mM) and 100 microliters of hydrogen peroxide. The absorbance was measured at 290 nm and presented as unit min^−1^ mg^−1^ protein.

### 4.9. Quantification of Endogenous IAA Levels

IAA was determined using the Salkowski reagent, as reported earlier [86]. Plant material (0.5 g) was crushed in 5 mL of distilled water using a mortar and pestle. For 15 min, the fluid was centrifuged at 10,000 rpm. In a test tube, 1 mL of the collected supernatant from the sample solution was combined with 2 mL of Salkowski reagent (Salkowski reagent = 1 mL of 0.5 M FeCl_3_ + 50 mL of 35% HClO_4_). The sample combination was incubated in the dark for 30 min at room temperature. The optical density was measured using a UV/VIZ spectrophotometer at 540 nm. As a control, 4 mL of Salkowski reagent was used.

### 4.10. Determination of Ions and Mineral Elements

To quantify ionic contents (K^+^, Ca^2+^, and Mg^2+^, Na^+^, and Cl^−^ ions), and mineral elements (nitrogen, phosphorus), about 0.5 g of powdered plant samples were digested with 6.5 mL of acid solution (HNO_3_, H_2_SO_4_, HClO_4_) in a 5:1:0.5 ratio, as described earlier [87]. The solution was heated until the appearance of white fumes, followed by the addition of distilled water and filtration using Whatman filter paper, No. 2. An iCE scientific 3000 atomic absorption spectrophotometer (Thermo Scientific, Waltham, MA, USA) was used to measure the ion concentrations in the digested solution, and hree biological replicates were tested for each element.

### 4.11. Statistical Analysis

GraphPad Prism 9.0.0 (121) software was used to statistically analyze the data sets that represent the means and standard errors of three independent replicates for each treatment. The Brown–Forsythe and Welch test were performed to run statistical evaluation through one-way analysis of variance (ANOVA). Significant differences (*p ≤* 0.05) are marked with asterisks in the presented data.

## 5. Conclusions

The current study found that BK inoculation in maize plants reversed the negative impact of salt stress by rebalancing the chlorophyll a/b ratio, carotenoids, total protein, total sugars, total lipids, secondary metabolites (phenol, flavonoids, tannins), antioxidant enzyme activity (catalase, ascorbate peroxidase), and osmoregulation (proline) to optimal levels. Furthermore, under salt stress, maize plants inoculated with BK showed decreased uptake and accumulation of Na^+^, and Cl^−^ ions, resulting in lower Na^+^/K^+^ and Na^+^/Ca^2+^ ratios, as well as greater N, P, Ca^2+^, K^+,^ and Mg^2+^ content than non-inoculated plants. The BK isolate increased salt tolerance in maize by modifying ionic root-to-shoot translocation, rebalancing Na^+^/K^+^, Na^+^/Ca^2+^, and scavenging ROS in maize plants under salt stress by inducing antioxidant enzymes to reach optimum levels. According to the current investigation, the association of BK increased maize performance in both normal and saline environments. As a result, it can be used as a growth booster for maize plants in salt-stressed environments.

## Figures and Tables

**Figure 1 plants-12-01703-f001:**
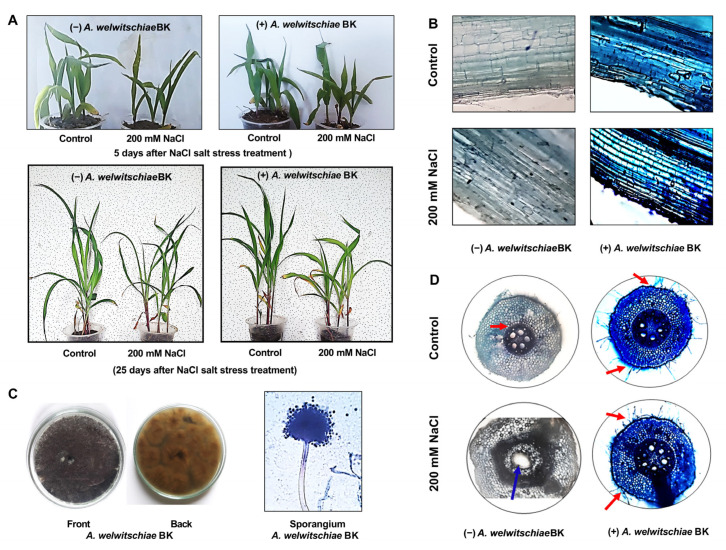
(**A**) Maize plant growth under control and salt stress conditions, both with *A. welwitschiae* BK supplementation. (**B**) Maize root transverse section showing endophytic colonization by *A. welwitschiae* BK, grown under salt stress conditions (200 mM NaCl) and normal conditions compared with the non-inoculated control. (**C**) Colony morphology of *A. welwitschiae* BK on PDA medium, front view (left penal), back view (middle penal), and sporangium morphology (right penal). (**D**) Maize root cross section showing the effect of *A. welwitschiae* BK on lysogenic aerenchyma production in maize roots under salt stress. Red arrows indicate colonization and the blue arrow indicates lysogenic aerenchyma spaces.

**Figure 2 plants-12-01703-f002:**
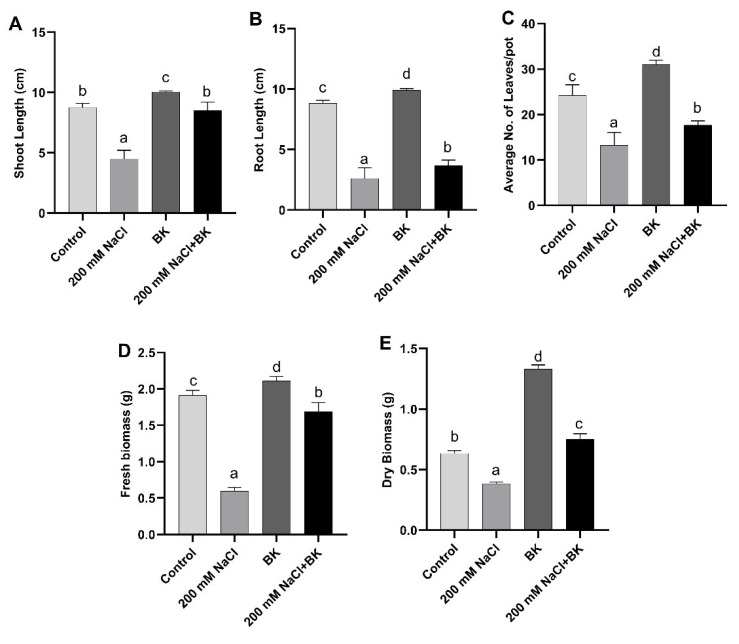
Influence of *A. welwitschiae* BK on vegetative growth parameters. (**A**) Shoot length, (**B**) root length, (**C**) average number of leaf/pots, (**D**) fresh biomass, and (**E**) dry biomass of maize grown under salt stress. Data represent the means and SE of three independent replicates for each treatment. Significant differences (*p ≤* 0.05) are marked with different letters.

**Figure 3 plants-12-01703-f003:**
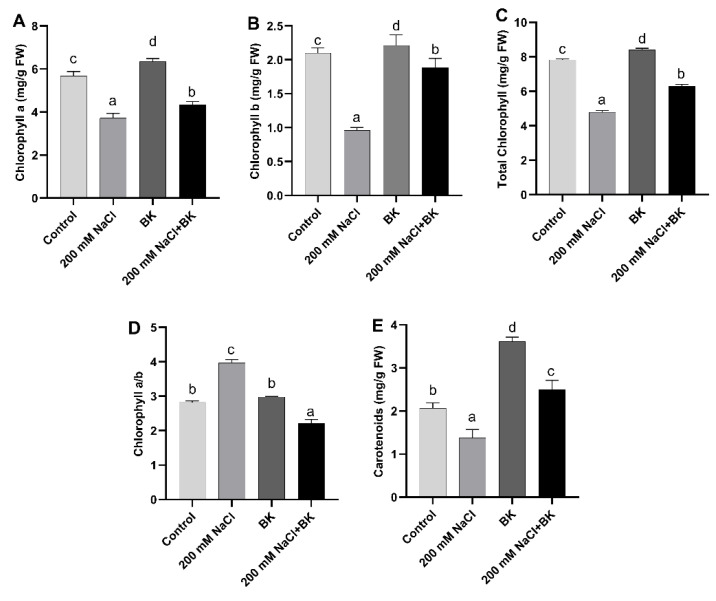
Influence of *A. welwitschiae* BK association on photosynthetic pigments. (**A**) Chlorophyll a, (**B**) chlorophyll b, (**C**) total chlorophyll, (**D**) chlorophyll a/b ratio, and (**E**) carotenoids of maize grown under salt stress. Data represent the means and SE of three independent replicates for each treatment. Significant differences (*p ≤* 0.05) are marked with different letters.

**Figure 4 plants-12-01703-f004:**
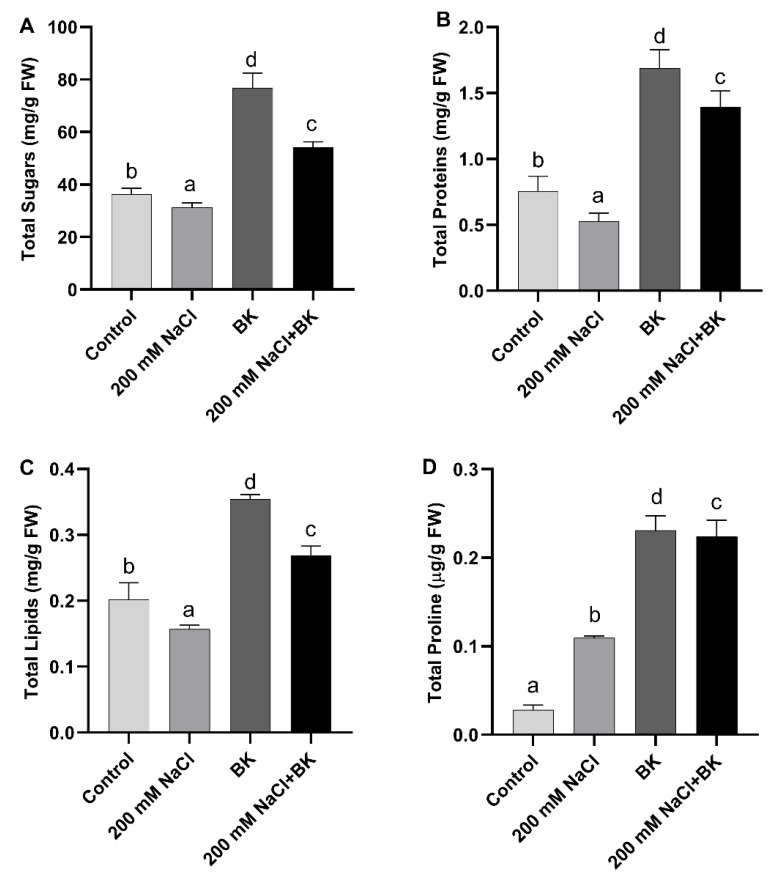
Influence of *A. welwitschiae* BK on primary metabolites. (**A**) Total soluble sugars, (**B**) total proteins, (**C**) total lipids, and (**D**) proline content of maize raised under salt stress. Data represent the means and SE of three independent replicates for each treatment. Significant differences (*p ≤* 0.05) are marked with different letters.

**Figure 5 plants-12-01703-f005:**
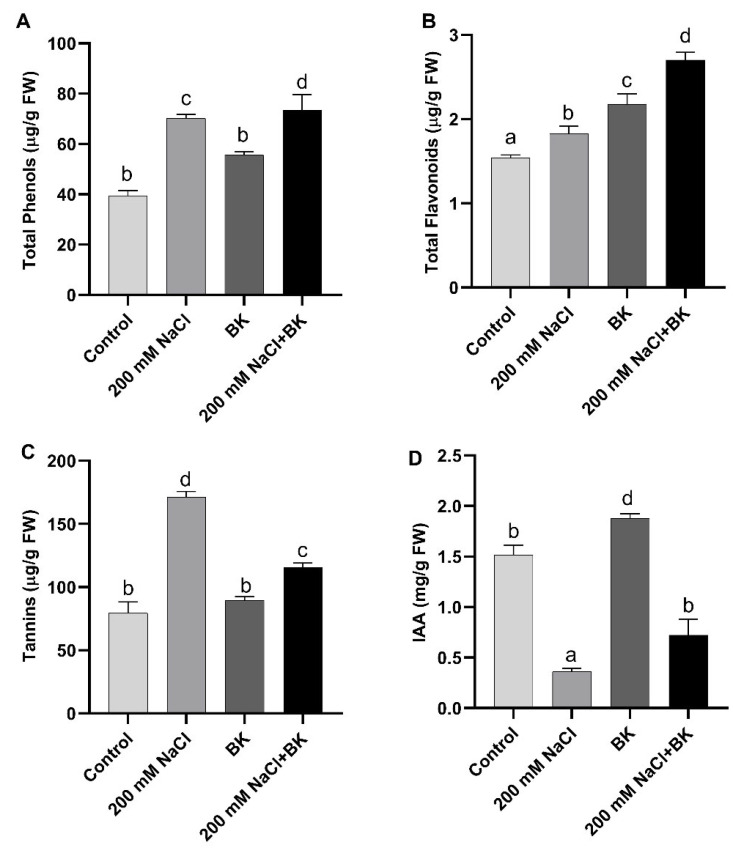
Influence of *A. welwitschiae* BK on secondary metabolites. (**A**) Total phenols, (**B**) total flavonoids, (**C**) tannin, and (**D**) IAA level of maize raised under salt stress. Data represent the means and SE of three independent replicates for each treatment. Significant differences (*p ≤* 0.05) are marked with different letters.

**Figure 6 plants-12-01703-f006:**
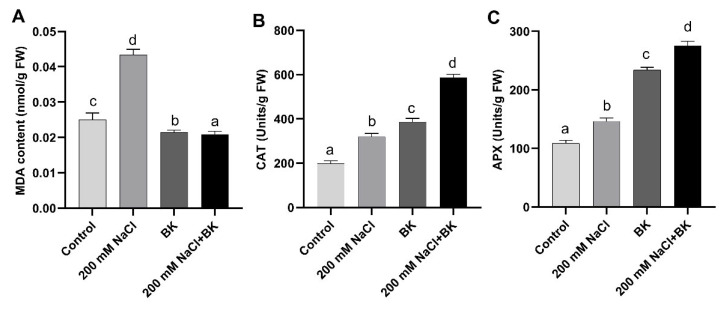
Influence of *A. welwitschiae* BK on antioxidant potential, (**A**) MDA content, (**B**) Catalase activity, (**C**) Ascorbate peroxidase activity of maize raised under salt stress. Data represent the means and SE of three independent replicates for each treatment. Significant differences (*p ≤* 0.05) are marked with different letters.

**Figure 7 plants-12-01703-f007:**
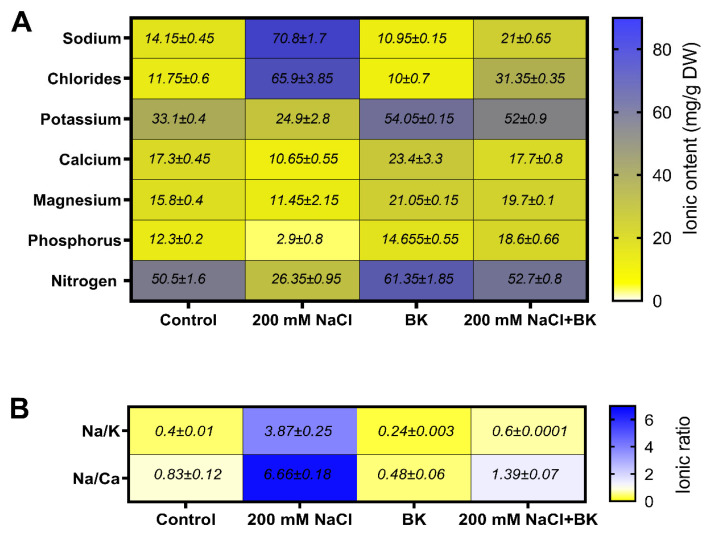
Influence of *A. welwitschiae* BK on the concentration of different ions. (**A**) Sodium, chloride, potassium, calcium, magnesium ions and nitrogen, phosphorus mineral elements (**B**) K^+^/Na^+^ and Na^+^/Ca^2+^ ion ratios in maize plants grown under salt stress. Data represent the means and SE of three independent replicates for each treatment. Significant data (*p ≤* 0.05) presents the mean ± SD.

**Table 1 plants-12-01703-t001:** The physical and chemical properties of the soil used in the experiment.

Characteristics	Values
Texture	Sandy-loam
Sand (%)	74.8
Silt (%)	11.9
Clay (%)	13.2
CEC (dS/cm)	4.3
ECe (dS/m)	0.9
pH	7.8
Carbonates (meq/L)	1.29
Bicarbonates (meq/L)	2.9
Chlorides (meq/L)	1.16
Organic matter (%)	1.3
Organic Carbon (%)	4.17

## Data Availability

The datasets presented in this article are not readily available because all the data are included in the manuscript. Requests to access the datasets should be directed to M.R., mamoona@awkum.edu.pk.

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
