# Peer review of "Aspergillus welwitschiae BK Isolate Ameliorates the Physicochemical Characteristics and Mineral Profile of Maize under Salt Stress"

_plants, 2023, doi:10.3390/plants12081703_

Round 1
Reviewer 1 Report
A paper on the effect of inoculation with a root endophyte on the resistance of maize to stress caused by salinity was reviewed. The work is clear in its design and in the presentation of the results. Some things can improve for it to be published.
There are some typographical errors throughout the document such as font size changes (444; 496-500), scientific names without typographical change according to previous sections (157), omission of letters in words (313), scientific notations according to the IS not correct ml versus mL which is correct as in line 261, lack of citations (472-474), omissions of points (689). There are also paragraphs marked in gray (444-449) and the citations section, some titles have errors in the spaces and italics in the scientific names, verify them well.
Related to the content, it is suggested that the photographs showing the evidence of endophytic colonization by A. welwitschiae must be of better quality and point out the aspects that change in the figure (321-322; 919-921). The photos have very poor resolution and it is not possible to glimpse the colonized area for those readers who have a high interest in seeing how to detect this symbiosis in the roots since there is very little documentation in this regard and this paper can be a good reference. If authors wanted to this publication must be really useful in terms of replicating the problem of salt stress alleviation with beneficial microorganisms in a crop that is globally necessary for humanity, please solve this observation well.
Regarding the number of leaves graphed in the maize plants, it is not clear that these averages are for the plants shown in the figure and in the cultivation time, they seem to be many, please review this.
Finally, the discussion is too long, it gives many theoretical elements from the literature as a justification (449 onwards) that really could be more succinct and put more case-citations of the changes that occurred in other studies since they are very poor. The references and the effects of salinity in cultivars are extensive and more so in maize. Please shorten the entire section and introduce better discussion on examples and less theoretical.
Author Response
POINT TO POINT RESPONSE
Endophytic Fungus (Aspergillus welwitschiae) Ameliorates the Physicochemical Characteristics and Mineral Profile of Maize Under Salt Stress
Reviewer 1
A paper on the effect of inoculation with a root endophyte on the resistance of maize to stress caused by salinity was reviewed. The work is clear in its design and in the presentation of the results. Some things can improve for it to be published.
Point 1:
There are some typographical errors throughout the document such as font size changes (444; 496-500), scientific names without typographical change according to previous sections (157), omission of letters in words (313), scientific notations according to the IS not correct ml versus mL which is correct as in line 261, lack of citations (472-474), omissions of points (689). There are also paragraphs marked in gray (444-449) and the citations section, some titles have errors in the spaces and italics in the scientific names, verify them well.
Response 1:
All these mentioned mistakes have been corrected as suggested by the reviewer. Corrections/Changes have been highlighted as RED text.
Point 2: Related to the content, it is suggested that the photographs showing the evidence of endophytic colonization by A. welwitschiae must be of better quality and point out the aspects that change in the figure (321-322; 919-921). The photos have very poor resolution and it is not possible to glimpse the colonized area for those readers who have a high interest in seeing how to detect this symbiosis in the roots since there is very little documentation in this regard and this paper can be a good reference. If authors wanted to this publication must be really useful in terms of replicating the problem of salt stress alleviation with beneficial microorganisms in a crop that is globally necessary for humanity, please solve this observation well.
Response 2:
All these mentioned points have been corrected and figure 1 is improved for more clarity and visibility of endophytic colonization with the roots of maize plants with and without salt stress. Colonization areas have been pointed with red arrows (see figure 1D). Also the text (321-322; 919-921) have been highlighted as RED.
Point 3: Regarding the number of leaves graphed in the maize plants, it is not clear that these averages are for the plants shown in the figure and in the cultivation time, they seem to be many, please review this.
Response 3:
Following statement has been mentioned in the manuscript (3.3 section, Line numbers 333-345).
“The growth performance of maize plants raised under salt stress, showed a significant (p<0.05) reduction in average of total number of leaves from 3 plants/pot compared to the unstressed control (C) maize plants. However, A. welwitschiae BK inoculation with maize under both the saline and normal conditions exhibited a significant (p<0.05) increase in examined parameter in comparison to respective control maze plants.”
Point 4.
Finally, the discussion is too long, it gives many theoretical elements from the literature as a justification (449 onwards) that really could be more succinct and put more case-citations of the changes that occurred in other studies since they are very poor. The references and the effects of salinity in cultivars are extensive and more so in maize. Please shorten the entire section and introduce better discussion on examples and less theoretical.
Response 4:
The discussion has been changed by removing unnecessary details, references, and theoretical elements.
Reviewer 2 Report
The authors in their manuscript entitled “Endophytic Fungus (Aspergillus welwitschiae) Ameliorates the Physicochemical Characteristics and Mineral Profile of Maize Under Salt Stress”, present the effect of Aspergillus welwitschiae (BK) on growth promotion of maize under severe salinity stress, after inoculation of the plants (via soil drenching) with a liquid spore suspension of the fungus.
The argument is clearly set, the methods applied are correct, results are well presented and adequately discussed. However, there are several questions that should be resolved through additional experimentation and further discussion, which relate to the experimental design and methodological approach. In specific:
1. The authors refer to the Aspergillus welwitschiae (BK) strain used as an endophyte. The strain origin is from another plant species in which it may grow as an endophyte. However, to conclude that the specific strain behaves as an endophyte in maize plants, it should be studied through the plant’s whole life cycle (for maize approx. 100-120 days) and conclude that it does not cause any disease symptom. This would cover the description of the specific strain as an endophyte. A. welwitschiae has been reported to cause Sisal Bole Rot, thus it does not always act as a beneficial or neutral endophyte, and it is generally reported as a main saprotroph. In the work presented the experiments performed end at the 35th day.
2. Another issue is that A. welwitschiae is also reported in literature as one of the main producers of Fumonisins. The authors thought they look to certain plant secondary metabolites, they do not test for FBs (e.g., FB2, FB4) production either in the fungal strain per se (in vitro) or their presence in planta. Fumonisins are major and dangerous contaminants, usually detected in corn due to Fusarium spp (Fumonisins producers) contamination. Thus, in combination to the previous comment regarding the plant’s whole life cycle it would be advisable to check for FBs production.
3. Another issue is that usually more than one strains are needed/used in this type of experimentation to prove that there is either the “x” benefit or negative effect. The use of at least another A. welwitschiae isolate (either endophyte isolated or an officially characterized strain) is/are needed as control or/and to demonstrate that there is a standard or/and contiguous effect. Particularly when the specific strain is suggested as a growth booster (lines 697-698).
4. A minor comment regards to measurement of other than IAA major salt stress associated hormones such as ABA and IBA.
Author Response
POINT TO POINT RESPONSE
Endophytic Fungus (Aspergillus welwitschiae) Ameliorates the Physicochemical Characteristics and Mineral Profile of Maize Under Salt Stress
Reviewer 2
Point 1.
The authors refer to the Aspergillus welwitschiae (BK) strain used as an endophyte. The strain origin is from another plant species in which it may grow as an endophyte. However, to conclude that the specific strain behaves as an endophyte in maize plants, it should be studied through the plant’s whole life cycle (for maize approx. 100-120 days) and conclude that it does not cause any disease symptom. This would cover the description of the specific strain as an endophyte. A. welwitschiae has been reported to cause Sisal Bole Rot, thus it does not always act as a beneficial or neutral endophyte, and it is generally reported as a main saprotroph. In the work presented the experiments performed end at the 35th day.
Response 1:
As suggested by the reviewer, the justifications and appropriate discussion have been included in the manuscript. Please see the line number 445-453.
“Recently, heavy metal tolerant endophytic fungi A. welwitschiae BK from Chlorophytum comosum has already been shown to improve the growth by ceasing the metal uptake and strengthening antioxidant system in Glycine max L. (Husna et al 2022). Since, A. welwitschiae BK was proved to be non-pathogenic on two host plants under normal and stressful environment, thereby, in the current study, non-pathogenic A. welwitschiae BK was exploited to mitigate the sodium salt stress tolerance in maize plants during initial growth and developmental phase, that is more sensitive towards environmental stresses such as salinity.”
- Another issue is that A. welwitschiae is also reported in literature as one of the main producers of Fumonisins. The authors thought they look to certain plant secondary metabolites, they do not test for FBs (e.g., FB2, FB4) production either in the fungal strain per se (in vitro) or their presence in planta. Fumonisins are major and dangerous contaminants, usually detected in corn due to Fusarium spp (Fumonisins producers) contamination. Thus, in combination to the previous comment regarding the plant’s whole life cycle it would be advisable to check for FBs production.
Response
Reviewer is right; however, researchers have also reported Variation in fumonisin and ochratoxin production associated with differences in biosynthetic gene content in Aspergillus niger and A. welwitschiae isolates from multiple crop and geographic origins. In addition to this, previous studies also indicate that a majority of A. niger isolates and only a minority of A. welwitschiae isolates can produce FBs. The relative abundance of each species and frequency of FBs -producing isolates can vary with crop and/or geographic origin (Susca, A., Proctor, R. H., Morelli, M., Haidukowski, M., Gallo, A., Logrieco, A. F., et al. (2016). Variation in fumonisin and ochratoxin production associated with differences in biosynthetic gene content in Aspergillus niger and A. welwitschiae isolates from multiple crop and geographic origins. Front. Microbiol. 7:1412. doi: 10.3389/fmicb.2016.01412).
Since in current study as well as in another report by Husan et al 2022, authors (researchers) have not found any symptoms that are related to the pathogenicity and toxicity for the plants (glycin max and maize) under experiment. Therefore, testing the FBs level in the of A. welwitschiae BK isolate is considered as out of the scope of aims and objective of the present study. However, in future of A. welwitschiae BK might be focus in relation to pathogenicity and mycotoxins related research, that might be another research project with appropriately defined aims and objective related to that aspect.
- Another issue is that usually more than one strains are needed/used in this type of experimentation to prove that there is either the “x” benefit or negative effect. The use of at least another A. welwitschiae isolate (either endophyte isolated or an officially characterized strain) is/are needed as control or/and to demonstrate that there is a standard or/and contiguous effect. Particularly when the specific strain is suggested as a growth booster (lines 697-698).
Response:
Thanks to the reviewer for the valuable suggestion. However, A. welwitschiae BK isolate is singly and newly isolated endophytic, nonpathogenic fungus. We do not have another A. welwitschiae isolate at the moment. However, it is valuable to fully characterize A. welwitschiae BK isolate in normal, abiotic ans well as biotic stress conditions in various plants. So that A. welwitschiae BK isolate can be used as a control for future studies to characterize further A. welwitschiae isolates, as suggested by reviewer.
- A minor comment regards to measurement of other than IAA major salt stress associated hormones such as ABA and IBA.
Response:
Keeping in view, the necessity of the salt stress associated hormonal response, the ABA production response along with other hormones in crosstalk will surely be evaluated in a separate research project, upon endophytic inoculation under normal and saline condition has been included in the manuscript. This is future prospect of authors to focus on the hormonal crosswalks and its modulation in maize upon salt stress during early (vegetative) and mature (reproductive) growth phases. This might be itself a sufficient amount to research work to be published for interested researchers. However, its future prospect of present study.
Currently, the main aim of present study was to evaluate the growth promotion and stress alleviation response driven by A. welwitschiae BK isolate in maize under salinity stress.
Reviewer 3 Report
The revised manuscript studies problematic and important problem of the impact of abiotic factors (here salinity) on crop production. Food production is key, especially nowadays when increasing population and environmental production forms constraints for sufficient and efficient food production. As there are vulnerable species for such factors there is a need for finding means for increasing their fitness. This manuscript shows one such natural solution - an inoculation with fungus showing resistance to salinity to provide maize protection to high concentration of salt.
For this reason the study is interesting, valuable and falls into the scope of the Journal.
It has informative title clearly indicating its content, good keywords and abstract which encourages for further reading. Introduction is very well constructed providing key information about the problems I mentioned above, describing maize and endophytes which may help plants coping with salt stress. This leads to clear aim of the study.
Experiment is well designed and all methods precisely described what make it easy to repeat it.
Results are clearly presented on 7 figures and well described. There are clear effects confirmed statistically.
Discussion is well structured explaining each effects of fungal inoculation on plant biology and biochemistry
All leads to convincing conclusions showing the positive role of the fungus - providing the potential tool for helping plants coping with salt stress.
Specific comments:
- for citation arabic numbers should be used instead of names
- the citation 'Kamran et al., 2018' (line 98) is not present in the references list
- L144, 152, 157, 312 - 'A. welwitschiae (BK)' should be also italized and not be presented in bold
- There are to Bradford (1976) references differing in page number. I would rather leave just one. Alternatively, please used (a) and (b) letters for differentiation
- line 224 - Khatiwora et al., (2017) - in the references this article dates on 2010
- line 271 - there is missing year after Lowry at al.
- line 285 - please correct VIZ to VIS
- in the references list Ali et al. 2022b is before 2022a. The references should be listed in the ascending order (a, b, c...)
- lines 445, 448-9, 496, 500 - the font size varies from the rest of the text
- line 475 - the citation of Ali et al. misses year of publication
- line 573 - there is a missing space in '...s (ROS)(Ozgur et al., 2013)...'
- line 655 - there is no need for upper index for 'outward'
- the reference Flagella Z., Cantore V., Giuliani M.M., Tarantino E., De Caro A. Crop salt tollerance: Physiological, yield and quality aspects. 784 Plant Biol. 2002;2:155–186 is not cites
- the reference Pereira, K., Sá, F., Torres, S. B., Paiva, E. P., Alves, T., & Oliveira, R. R. (2021). Exogenous application of organic acids in maize 859 seedlings under salt stress. Brazilian journal of biology = Revista brasleira de biologia, 84, e250727. https://doi.org/10.1590/1519- 860 6984.250727 is not cited
- the reference Sun, Y., Mu, C., Zheng, H. et al. Exogenous Pi supplementation improved the salt tolerance of maize (Zea mays L.) by promoting 891 Na+ exclusion. Sci Rep 8, 16203 (2018). https://doi.org/10.1038/s41598-018-34320-y is not cited
Author Response
POINT TO POINT RESPONSE
Endophytic Fungus (Aspergillus welwitschiae) Ameliorates the Physicochemical Characteristics and Mineral Profile of Maize Under Salt Stress
Reviewer 3
The revised manuscript studies problematic and important problem of the impact of abiotic factors (here salinity) on crop production. Food production is key, especially nowadays when increasing population and environmental production forms constraints for sufficient and efficient food production. As there are vulnerable species for such factors there is a need for finding means for increasing their fitness. This manuscript shows one such natural solution - an inoculation with fungus showing resistance to salinity to provide maize protection to high concentration of salt.
For this reason the study is interesting, valuable and falls into the scope of the Journal.
It has informative title clearly indicating its content, good keywords and abstract which encourages for further reading. Introduction is very well constructed providing key information about the problems I mentioned above, describing maize and endophytes which may help plants coping with salt stress. This leads to clear aim of the study.
Experiment is well designed and all methods precisely described what make it easy to repeat it.
Results are clearly presented on 7 figures and well described. There are clear effects confirmed statistically.
Discussion is well structured explaining each effects of fungal inoculation on plant biology and biochemistry
All leads to convincing conclusions showing the positive role of the fungus - providing the potential tool for helping plants coping with salt stress.
Specific comments:
- for citation arabic numbers should be used instead of names
Response:
Citation will be converted to the formate of journal, as soon as manuscript is ready for formatting.
- the citation 'Kamran et al., 2018' (line 98) is not present in the references list
Response:
'Kamran et al., 2018' reference has been added at reference line number 25.
- L144, 152, 157, 312 - 'A. welwitschiae (BK)' should be also italized and not be presented in bold
Response:
'A. welwitschiae (BK)' name correction has been done throughout the manuscript.
- There are to Bradford (1976) references differing in page number. I would rather leave just one. Alternatively, please used (a) and (b) letters for differentiation
Response:
Correction has been followed.
- line 224 - Khatiwora et al., (2017) - in the references this article dates on 2010
Response:
Correction has been done.
- line 271 - there is missing year after Lowry at al.
Response:
Correction has been followed
- line 285 - please correct VIZ to VIS
Response:
- in the references list Ali et al. 2022b is before 2022a. The references should be listed in the ascending order (a, b, c...)
Response:
Correction has been done
- lines 445, 448-9, 496, 500 - the font size varies from the rest of the text
Response:
Correction has been done
- line 475 - the citation of Ali et al. misses year of publication
Response:
Correction has been done
- line 573 - there is a missing space in '...s (ROS)(Ozgur et al., 2013)...'
Response:
Correction has been done
- line 655 - there is no need for upper index for 'outward'
Response:
Correction has been done
- the reference Flagella Z., Cantore V., Giuliani M.M., Tarantino E., De Caro A. Crop salt tollerance: Physiological, yield and quality aspects. 784 Plant Biol. 2002;2:155–186 is not cites
Response:
Correction has been done. Unnecessary refrence hs been removed.
- the reference Pereira, K., Sá, F., Torres, S. B., Paiva, E. P., Alves, T., & Oliveira, R. R. (2021). Exogenous application of organic acids in maize 859 seedlings under salt stress. Brazilian journal of biology = Revista brasleira de biologia, 84, e250727. https://doi.org/10.1590/1519- 860 6984.250727 is not cited
Response:
Correction has been done. Unnecessary refrence hs been removed.
- the reference Sun, Y., Mu, C., Zheng, H. et al. Exogenous Pi supplementation improved the salt tolerance of maize (Zea mays L.) by promoting 891 Na+ exclusion. Sci Rep 8, 16203 (2018). https://doi.org/10.1038/s41598-018-34320-y is not cited
Response:
Correction has been done. Citation has been incorporated.
Reviewer 4 Report
Interesting results were obtained, but the scientific analysis, data description and scientific style need to be significantly improved.
The title is misleading, as Aspergillus welweitschiae is not an endophyte of Zea mays, but has been an endophyte of Chlorophytum comosum as it was isolated from that plant.
One of the main problems both in Introduction and Discussion is lack of references in sentences where factual information is given. Each scientific fact needs to be traceable to the paper where it has been primary established (experimental paper) or to review paper where the particular relationship has been described. I include a list of lines where sentences are starting, which need to have proper references included: 35, 40, 42, 44, 48, 62, 74, 76, 94, 100, 103, 109, 110, 112, 127, 449, 452, 455, 457, 458, 463, 470, 472, 481, 483, 487, 488, 490, 502, 508, 509, 517, 518, 521, 522, 527, 531, 533, 546, 553, 554, 556, 557,565, 568, 573, 574, 576, 577, 582, 586, 592, 597, 598, 608, 613, 621, 624, 627, 637, 645, 547, 649, 656, 658, 662, 664, 667, 669, 671, 675, 676, 678.
It is very strange that in all places where Aspergillus welwitschiae are mentioned, it is followed by "BK" in parentheses. If this is meant to be a designation of a particular strain, then it needs to be defined in Materials and methods, and further only short name has to be used (A. welwitschiae).
Introduction
Line 48, this idea is not factually true.
Line 69, "miaze". This sentence is dubious, as all plants have evolved the mentioned enzymes.
Line 72, what is meant by "phytohormones with crosstalk also participates"?
Line 92, "IAA" not given in full.
Line 94, IAA and SA cannot be called "these acids". Both are established as plant hormones, not "chemical messengers".
Line 96, "exogenous supplementation of exogenous Pi", and what is Pi?
Line 106, "Plant-microbe interactions" include also pathogens.
Line 140, do not include conclusions.
Materials and methods
In Materials and methods, use short forms of headings avoiding lengthy repetitions of "... in maize plants inoculated with A. welwitschiae (BK) under salt stress".
Description of all performed actions need to be clear and straightforward. Now it is rather chaotic with many repetitions, difficult to follow. For example, why under 2.3. it is claimed that "After examining plant growth and biochemistry, an experiment was conducted to examine ..."?
It is strange that no concentration of plant-available nutrients are reported in Table 1. If no fertilization was provided during the experiment, it is highly possible that plants were in a nutrient-deprived state.
Line 172, it is claimed that maize "has shown stunted growth up to 200 mM NaCl" with reference to Chen et al. 2007. Most likely, it is meant "starting from 200 mM NaCl. However, in publication of Chen et al., 200 mM NaCl was used for irrigation purposes within 16 days at 3 days intervals, and then plants died. In the present study, only 2 mL of 200mM NaCl was applied to each pot twice a week for unspecified period of time. It is strangely mentioned that only "5-day-old maize seedlings" were treated, which cannot be true. So, this treatment cannot be compared to the one described by Chen et al. 2007.
Do not include sentences clearly belonging to literature analysis (starting in lines 172, 246.
Description in 2.4. is incomplete, as a part on calculation by specific formulae is missing.
In 2.8, all paragraph on MDA analysis and part of paragraph on enzyme assays are directly copied from Raja et al. (2020) 3 Biotech 10: 208. Even the name of the wrong model organism ("2-month-old tomato plants") has been left. Surprisingly, Raja et al. correctly indicates that APX has been extracted in a buffer with ascorbic acid added, but this part is missing in the current manuscript. However, if no ascorbic acid was added, this means that the obtained data on APX activity simply cannot be used, as this enzyme looses activity if extracted without ascorbic acid. Also, it appears now that two different methods have been used for protein quantification.
Line 280, it is more likely not "crusher" but "mortar".
In 2.10. nitrogen and phosphorus do not represent ions, use of "mineral elements" would be more correct.
Results
Figures and tables need to be incorporated within the main body of the manuscript.
Figure 1A cannot be used in the present form, as it contains two parts with clearly different scales, with no scale bars. Photographs in Figure 1B have no scientific relevance. Also, no much relevance for root sections, as no scale bars or other markings have been provided.
In Figures 2 to 6, provide indication of statistical significance of differences between all treatments by using different letters as in your previous papers.
In Figure 7, provide numeric values for measurements in a form of table instead of color scale, together with indication of statistical significance of differences among all treatments.
The results itself are described in a sloppy manner, simply repeating all numeric values from figures. Instead, main effects need to be described in a comparative way, excluding repetition of the numeric values.
Not enough detail are given in figure legends to understand performed measurements. First of all, it is the effect by both NaCl and the microorganism.
In 3.10., nitrogen and phosphorus do not represent ions.
Discussion
Discussion is rather general and too long. Many sentences are just general, introduction-level texts, with no proper references for factual claims. It is necessary to concentration on most important mechanisms leading to increase of salt tolerance of Zea mays by Aspergillus welwitschiae and experiments in similar model systems.
Line 442, "of salt tolerant of fungal endophytes". Besides this, no salt tolerance of Aspergillus welwitschiae has been reported.
References in the text and list of references are not according to the style adopted by the journal. It is full of style inaccuracies.
Author Response
Reviewer 4
Interesting results were obtained, but the scientific analysis, data description and scientific style need to be significantly improved.
The title is misleading, as Aspergillus welweitschiae is not an endophyte of Zea mays, but has been an endophyte of Chlorophytum comosum as it was isolated from that plant.
As suggested by the reviewer, the title has been changed to “Aspergillus welwitschiae BK Ameliorates the Physicochemical Characteristics and Mineral Profile of Maize Under Salt Stress”
One of the main problems both in Introduction and Discussion is lack of references in sentences where factual information is given. Each scientific fact needs to be traceable to the paper where it has been primary established (experimental paper) or to review paper where the particular relationship has been described. I include a list of lines where sentences are starting, which need to have proper references included: 35, 40, 42, 44, 48, 62, 74, 76, 94, 100, 103, 109, 110, 112, 127, 449, 452, 455, 457, 458, 463, 470, 472, 481, 483, 487, 488, 490, 502, 508, 509, 517, 518, 521, 522, 527, 531, 533, 546, 553, 554, 556, 557,565, 568, 573, 574, 576, 577, 582, 586, 592, 597, 598, 608, 613, 621, 624, 627, 637, 645, 547, 649, 656, 658, 662, 664, 667, 669, 671, 675, 676, 678.
Response:
Whole introduction and discussion part has been reorganized and rewritten. Updated, references have been incorporated correctly and irrelevant references have been removed.
All references have been connected to the scientific facts and in closed conneteion to the current study, for proper comparisons.
It is very strange that in all places where Aspergillus welwitschiae are mentioned, it is followed by "BK" in parentheses. If this is meant to be a designation of a particular strain, then it needs to be defined in Materials and methods, and further only short name has to be used (A. welwitschiae).
Response:
- welwitschiae BK name has been corrected in method section and throughout the manuscript, consistently.
Introduction
Line 48, this idea is not factually true.
Response:
Correction has been followed.
Line 69, "miaze". This sentence is dubious, as all plants have evolved the mentioned enzymes.
Response:
Correction has been followed.
Line 72, what is meant by "phytohormones with crosstalk also participates"?
Response:
Correction has been followed, and ambiguity is removed.
Line 92, "IAA" not given in full.
Response:
Correction has been followed.
Line 94, IAA and SA cannot be called "these acids". Both are established as plant hormones, not "chemical messengers".
Response:
Correction has been followed.
Line 96, "exogenous supplementation of exogenous Pi", and what is Pi?
Response:
Correction has been followed. Pi (H2PO4-) is incorporated.
Line 106, "Plant-microbe interactions" include also pathogens.
Response:
Correction has been followed.
Line 140, do not include conclusions.
Response:
Correction has been followed.
Materials and methods
In Materials and methods, use short forms of headings avoiding lengthy repetitions of "... in maize plants inoculated with A. welwitschiae (BK) under salt stress".
Response:
Correction has been followed.
Description of all performed actions need to be clear and straightforward. Now it is rather chaotic with many repetitions, difficult to follow. For example, why under 2.3. it is claimed that "After examining plant growth and biochemistry, an experiment was conducted to examine ..."?
Response:
Correction has been followed.
It is strange that no concentration of plant-available nutrients are reported in Table 1. If no fertilization was provided during the experiment, it is highly possible that plants were in a nutrient-deprived state.
Response:
Information has been added in method section.
Line 172, it is claimed that maize "has shown stunted growth up to 200 mM NaCl" with reference to Chen et al. 2007. Most likely, it is meant "starting from 200 mM NaCl. However, in publication of Chen et al., 200 mM NaCl was used for irrigation purposes within 16 days at 3 days intervals, and then plants died. In the present study, only 2 mL of 200mM NaCl was applied to each pot twice a week for unspecified period of time. It is strangely mentioned that only "5-day-old maize seedlings" were treated, which cannot be true. So, this treatment cannot be compared to the one described by Chen et al. 2007.
Response:
Information has been added in method section. Method has been written clearly to remove ambiguity.
Do not include sentences clearly belonging to literature analysis (starting in lines 172, 246.
Description in 2.4. is incomplete, as a part on calculation by specific formulae is missing.
In 2.8, all paragraph on MDA analysis and part of paragraph on enzyme assays are directly copied from Raja et al. (2020) 3 Biotech 10: 208. Even the name of the wrong model organism ("2-month-old tomato plants") has been left. Surprisingly, Raja et al. correctly indicates that APX has been extracted in a buffer with ascorbic acid added, but this part is missing in the current manuscript. However, if no ascorbic acid was added, this means that the obtained data on APX activity simply cannot be used, as this enzyme looses activity if extracted without ascorbic acid. Also, it appears now that two different methods have been used for protein quantification.
Response:
Typographical mistakes have been removed from method section. Correct refrene has been ncorporated.
Line 280, it is more likely not "crusher" but "mortar".
Response:
Correction has been done.
In 2.10. nitrogen and phosphorus do not represent ions, use of "mineral elements" would be more correct.
Response:
Correction is done in method, result and discussion section.
Results
Figures and tables need to be incorporated within the main body of the manuscript.
Figure 1A cannot be used in the present form, as it contains two parts with clearly different scales, with no scale bars. Photographs in Figure 1B have no scientific relevance. Also, no much relevance for root sections, as no scale bars or other markings have been provided.
Response:
Figure 1 has been improved. Indicators have been mentioned with sufficient details.
In Figures 2 to 6, provide indication of statistical significance of differences between all treatments by using different letters as in your previous papers.
Response:
Figure 2 to 6 has been changed with regards to the significant letters instead of the asterisk.
Asterisk have been removed from all figures, as suggested by the reviewer.
In Figure 7, provide numeric values for measurements in a form of table instead of color scale, together with indication of statistical significance of differences among all treatments.
Response:
As suggested by the reviewer, figure 7 is provided with the numeric values for measurements in the form of heat mapper with color scale, together with indication of statistical significance of differences among all treatments. Data has been shown as mean with SD.
The results itself are described in a sloppy manner, simply repeating all numeric values from figures. Instead, main effects need to be described in a comparative way, excluding repetition of the numeric values.
Response;
Repetition has been removed.
Not enough detail are given in figure legends to understand performed measurements. First of all, it is the effect by both NaCl and the microorganism.
Response:
Information have been incorporated in legends for more clarity.
In 3.10., nitrogen and phosphorus do not represent ions.
Response:
Correction has been done as mineral elements.
Discussion
Discussion is rather general and too long. Many sentences are just general, introduction-level texts, with no proper references for factual claims. It is necessary to concentration on most important mechanisms leading to increase of salt tolerance of Zea mays by Aspergillus welwitschiae and experiments in similar model systems.
Line 442, "of salt tolerant of fungal endophytes". Besides this, no salt tolerance of Aspergillus welwitschiae has been reported.
Response:
Discussion has been mainly rewritten with incorporation of new relevant references.
References in the text and list of references are not according to the style adopted by the journal. It is full of style inaccuracies.
Response:
All references have been arranged according to the ascending order along the main text.
Formatting with numbers will be done.
Thank you so much for your time.
Best regards
Reviewer 5 Report
Comments on the ms entitled “Endophytic Fungus (Aspergillus welwitschiae) Ameliorates the Physicochemical Characteristics and Mineral Profile of Maize Under Salt Stress”
The reviewed work deals with an important aspect of food production which is the identification and subsequent use of the endophytic fungus. In my opinion, the Authors of the work have achieved their goals, namely: explore the mechanism behind the positive influence driven by A. welwitschiae (BK) in maize upon salt stress and recognize possible factors inducting the salt stress tolerance in maize upon association with selected endophytic fungus.
Undertaking research on this topic is an important aspect of this work. The results obtained are interesting and should be continued.
I have some comments regarding the layout of the paper and the presentation of the research results, which I have included in the text of the article.

Author Response
Reviewer 5
The reviewed work deals with an important aspect of food production which is the identification and subsequent use of the endophytic fungus. In my opinion, the Authors of the work have achieved their goals, namely: explore the mechanism behind the positive influence driven by A. welwitschiae (BK) in maize upon salt stress and recognize possible factors inducting the salt stress tolerance in maize upon association with selected endophytic fungus.
Undertaking research on this topic is an important aspect of this work. The results obtained are interesting and should be continued.
I have some comments regarding the layout of the paper and the presentation of the research results, which I have included in the text of the article.
Response:
All the changes and corrections have been done as suggested by the reviewer.
Best regards
Round 2
Reviewer 2 Report
The fact that the authors state that they have no other A. welwitschiiae isolate/strain available does not mean that they could not use/order a certified strain available from a repository.
It still is a main drawback in this research.
Reviewer 4 Report
Thank you for corrections
Reviewer 5 Report
Comments on the ms plants-2133872: "Endophytic Fungus (Aspergillus welwitschiae) Ameliorates the Physicochemical Characteristics and Mineral Profile of Maize Under Salt Stress".
Dear Authors,
I appreciate the corrections and additions made to the manuscript. I believe that the paper should be published in the journal Plants in its present form.
Sincerely